# Study and Characterization of the Dielectric Behavior of Low Linear Density Polyethylene Composites Mixed with Ground Tire Rubber Particles

**DOI:** 10.3390/polym12051075

**Published:** 2020-05-08

**Authors:** Marc Marín-Genescà, Jordi García-Amorós, Ramon Mujal-Rosas, Lluís Massagués, Xavier Colom

**Affiliations:** 1Department of Mechanical Engineering, ETSEQ-URV, Països Catalans, 26, 45002 Tarragona, Spain; 2Department of Electrical Engineering, ETSE-URV, Països Catalans, 26, 45002 Tarragona, Spain; jordi.garcia-amoros@urv.cat (J.G.-A.); lluis.massagues@urv.cat (L.M.); 3Department of Electrical Engineering, ESEIAAT-UPC, Colom, 1, 08222 Terrassa, Spain; mujal@ee.upc.edu; 4Department of Chemical Engineering, ESEIAAT-UPC, Colom, 1, 08222 Terrassa, Spain; xavier.colom@upc.edu

**Keywords:** reused tires, LLDPE, electrical properties, electrical modulus, composite recycling applications

## Abstract

The waste rubber vulcanizate, on account of its stable, cross-linked and three-dimensional structural arrangement, is difficult to biodegrade. Thus, the ever-increasing bulk of worn-out tires is a serious environmental issue and its safe disposal is still a challenging task reported widely by the scientific community. The rubber materials, once they end their useful life, may present difficulties to be reused or recycled. At present, only one tire recycling method is used, which involves grinding and separating steel and fibers from vulcanized rubber, and then using rubber for industrial applications, such as flooring, insulation, footwear. In this paper, a new compound material is presented from a base of reused tire powder (Ground Tire Rubber: GTR) as a mixer and linear low-density polyethylene (LLDPE) as a matrix. The reused tire powder, resulting from grinding industrial processes, is separated by sieving into just one category of particle size (<200 μm) and mixed with the LLDPE in different amounts (0%, 5%, 10%, 20%, 40%, 50% and 70% GTR). Due to the good electrical properties of the LLDPE, this study’s focus is settled on the electrical behavior of the obtained composites. The test of the dielectric behavior is carried out by means of DEA test (Dynamic Electric Analysis), undertaken at a range of temperatures varying from 30 to 120 °C, and with a range of frequencies from 1 to 10^2^, to 3·10^6^ Hz, from which permittivity, conductivity, dielectric constant and electric modulus have been obtained. From these experimental results and their analysis, it can be drawn that the additions of different quantities of GTR to LLDPE could be used as industrial applications, such as universal electrical cable joint, filler for electrical applications or cable tray systems and cable ladder system.

## 1. Introduction

Tire rubber wastes are a serious world environment problem and its safe disposal is still a challenging task reported widely by the scientific community [1,2], the main cause is due to the difficulties in rubber waste recycled, that causes harmful environmental effects. Tires containing almost 50% rubber are polymeric materials. The global production of rubber materials in last years was about nearly 27–30 MTons [1]. However, the tire industries, as the main application of rubbers (65% of the global rubber production), generate the largest amounts of tire waste materials. Because of this, 1.5 billion waste tires per year are discarded worldwide, containing up to 90% of vulcanized rubber that cannot be easily recycled (reprocessed) due to its crosslinked structure [3,4]; therefore, the management and recycle of used tires has become such a huge environmental challenge. Likewise, many efforts are being made to find new fields of application that can absorb the large amount of waste tire rubber that is generated yearly. The use of these materials as reinforcements or blends in composite materials has been widely studied in many works [5,6,7], but the presence of these in composites of polymeric matrix modifies physical characteristics, for instance, dielectric, mechanical, thermal behavior. Vulcanized waste rubbers are difficult to recycle, due to the crosslink structure of rubber [8]. Therefore, an interesting option is to blend waste tires with plastics to decrease the final costs of the products due to a lower amount of virgin material being used. Waste tires need to be shredded into smaller particles (microparticles) for easier addition into plastic matrices, and the separation of textiles and steels from waste tires is required [9]. Shredded tires can be used in virgin/fresh polymers such as rubbers, thermoplastics, and thermoset blends for civil engineering, automotive applications, and other uses. Therefore, blends of rubber with thermoplastics are consuming a large amount of waste tires [10,11]. The addition of GTR particles into polymeric thermoplastic matrices has proven to be promising in terms of cheaper product, a cleaner environment and viable processing. Polymer blends’ recycling of waste tires seems a promising field of studies due to balanced properties obtained by partially replacing the virgin polymer with waste tires (GTR) [12,13]; so, actual research experiences in the rubber recycling as a polymeric reinforcement suggested that this field is hopefully being explored by the scientific community. 

Studies reveal that the recycling process is an excellent option to treat waste polymer products in comparison with the old-style methods (landfill accumulation, combustion of waste polymers, burying underground) that lead to negative impacts on the environment [14,15], and this trend is confirmed by the increasing of the recycling yield rate and the weight of recycled materials, which was gradually increased to more than 70% [16]. Thermoplastic blends are useful for developing new recycling strategies and developing new composites materials after value addition. Moreover, different actual research experiences develops recycling thermoplastic blends can be mixed up together for making blends in order to produce further value-added products since those have comparatively similar properties [17]; so, the development of more recycling composite applications is a necessary research activity in order to evaluate the feasibility of the recycling composites. 

In the present research, the matrix is the thermoplastic polymer (linear low-density polyethylene (LLDPE)) and the reinforcements are GTR particles. Significant works studied the mechanical and morphological properties of the GTR blends [18,19,20,21]. Despite the polymeric blends with GTR behavior showing a low compatibility between both phases (polymeric/GTR loads) [22,23], the cheaper and easier process of blended GTR into thermoplastic matrices gives an opportunity to this GTR recycling strategy. With this aim, this article looks for what is the ideal amount of GTR loads in polymeric matrices, for different industrial applications analyzed in this paper. For this reason, in the actual research are the compared results from the Spanish standards (UNE) and International standards (IEC) specific applications, used as an application standard in industrial developments with dielectric results obtained to study the industrial application feasibility of the composites analyzed according to specific standards application. This analysis conducts the classification of some low-requirement applications in the industrial field, whereby such applications can be applied to this composite (LLDPE + GTR), for a low GTR content. The analyzed applications will be industrial applications, such as universal electrical cable joint, filler for electrical applications and cable tray systems and cable ladder systems.

The composite materials are heterogeneous, and their properties depend on the quantity, size and shape of the reinforcement, as well as other factors, such as their preparation, as well as compatibility. One way to increase the compatibility between components is to reduce the degree of cross-linked GTR by devulcanization [24,25]. Relevant changes in the properties are observed when changes to the size of the reinforcement particles take place. In fact, previous investigations [26,27] have shown that there are better mechanical properties for samples with particle diameter under 200 µm. Thus, this research is focused on particles of GTR with a diameter under 200 µm. For this purpose, it was used as a matrix LLDPE, which has good electrical, processing and mechanical properties as well as relatively low price [28,29,30]. In general, the incorporation of fillers in LLDPE increases the elastic modulus of the material and its tensile strength, but often decreases the elongation at break [31,32].

The complete dielectric characterization behavior of thermoplastic polymer LLDPE with the additions of different GTR particle amounts, as a function of frequency and temperature, is provided. Is expected that carbon black (CB) in the GTR particles may affect the conductive process in the polymeric composites (LLDPE + GTR) [33], affecting electrical insulation properties. Other research [34,35,36] studied the dielectric properties of composite polymers, depending on frequency and temperature. The Maxwell–Wagner–Sillars (MWS) interface polarization theory is observed in heterogeneous systems composed of two or more phases. In polymer composites, interface polarization is almost always present, because additives or impurities make them heterogeneous systems. In general, in systems with a conductive component, the dielectric’s permittivity, relaxation interfaces and conductivity can be both high and low frequencies. To overcome this difficulty in the study of interface polarization, the electric module is used in polymers and composites to study its conductivity relaxation behavior [37,38,39,40]. 

### Featured Application

The aims of this manuscript are to analyze the dielectric behavior of different seven composite materials obtained by mixing LLDPE polymer with different percentages of GTR (up to 70%) to see their response as function of the amount of GTR particles and its possible feasibility to be used in specific applications in the industrial electric field. Therefore, the double objective of this research is framed in the characterization and study of the electrical properties of composite materials, the evaluation and characterization of the electrical behavior of LLDPE + GTR compounds, and the use of some of these compounds for the industry, mainly for insulating, in order to give output to recycle materials coming from tires that are out of use and to give a possible industrial application, purposing a possible application for the recycling of tire rubbers. 

## 2. Materials and Methods

### 2.1. Materials

Linear low-density polyethylene (LLDPE) was supplied by Montachem International (Fort Lauderdale, FL, USA). The physical and tensile properties of this polymeric material are summarized in Table 1. On the other hand, the old used tire (GTR), with a particle size less than 200 μm, has verified, by TGA analysis (thermogravimetric analysis), that carbon black content was about 35%. The original GTR was separated by sieving into one size particles: <200 μm.

#### 2.1.1. Preparation of the Compound

The recycled tire powder was dried in an oven at 100 °C for 24 h. Samples of Polymer/GTR compound, varying the composition (0%, 5%, 10%, 20%, 40%, 50% and 70% of GTR), were prepared. The mixing process was done with a Brabender mixer machine at processing temperature, and, to prevent the degradation of the polymer, the mixing time was limited to 4 min. After the mixing process, all the samples were homogenized in a laboratory two roll mill (Collin W100T, Collin, Maitenbeth, Germany) at 60 °C to improve the dispersion of GTR particles. LLDPE/GTR laminates were obtained by using a hot plate press (Collin, Maitenbeth, Germany) at 100 kN and a pressing temperature for 10 min (Table 1). The cooling stage was done with a closed water circuit, which was held in the same press and pressure for 5 min. Samples for testing were properly set up according to the specifications of ASTM D-150. A sample of the pure LLDPE polymer was also prepared with the same requirements in order to obtain comparable results. One specimen for testing was used.

#### 2.1.2. Dielectric Analysis

The dielectric analysis was performed with GTR particles smaller than 200 μm. The dielectric parameters and magnitudes were measured by means of the Dynamic Electric Analysis (DEA) test [41] (also called Dielectric Analysis) with BDS40 equipment (Figure 1, Montabaur, Germany) in which a temperature sensor was incorporated using a compression mold of 2.5 cm in diameter and 0.1 mm thick. The measurements were carried out in a frequency range between 10^−2^ and 3 × 10^6^ Hz, with a temperature sweep between 30, 100 and 120 °C and at a speed of 3 °C/min.

#### 2.1.3. Samples

There are two types of specimens: one obtained by neat matrix (LLDPE) and the other type of specimen is containing the blended LLDPE polymer with different amounts of GTR particles in the mixers. The dimensions of the specimens are defined by ASTM D-150 [42,43,44] and are shown in Figure 2a. This Figure shows that the specimens are cylindrical 2.5 mm in diameter and 0.1 mm thick. In Figure 2b, blended LLDPE + GTR samples can be seen between the two electrodes used in the test. Once the specimen is placed between the two electrodes, it must be introduced into a test chamber to provide the desired temperature during the experiment time. The test system then carries out the measurements for the different frequencies and temperatures configured. The software used to obtain the data is WinDETA 4.1 (Montabaur, Germany). This was used to control the DEA system in order to configure the test of the different electrical properties to be obtained and analyzed.

## 3. Microstructural and Electrical Characterization Results

### 3.1. Microstructural Analysis of the Compounds LLDPE/GTR

Different microphotographs of the interphase GTR–LLDPE matrix were performed using Scanning Electron Microscopy (SEM) with different magnifications. SEM was used to analyze the fracture surface of those broken samples in tensile strength tests. It is possible to analyze the effects of GTR inside the matrix by observing the environment of the reinforcement particles. A JEOL 5610 microscope (Tokyo, Japan) was used, and the samples were previously coated with a thin layer of gold to increase conductivity.

GTR particles do not reach the degradation temperature when mixing, and it is possible to see those particles dispersed in the homogeneous LLDPE polymer matrix. As shown Figure 3a, the compatibility between GTR and LLDPE is not enough good. Moreover, the gap between components damages the electrical properties. Specifically, the SEM photographs show a compound with GTR, which causes low interfacial adhesion. So, as a result, with high GTR concentrations there is a greater potential for particle agglomeration, which obstructs interfacial adhesion. The percentage of LLDPE is not enough to wrap the GTR particles, making bonding more difficult, with cracks and pores of considerable sizes appearing in their borders. The interaction between the matrix and reinforcement particles of GTR is very low. As depicted Figure 3b, the degree of dispersion improves, avoiding the formation of clusters when the amount of GTR is lower to 20%. This picture shows that small particles are wrapping for the LLDPE matrix, although in the central part of picture appears a big particle (150 μm of length) that explains the lack of interaction between both components. These results agree with the tensile properties obtained in a manuscript published by Yao Du et al. [45].

### 3.2. Conductivity

Figure 4a displays the conductivity results of the different compounds LLDPE/GTR at 30 °C. The neat LLDPE conductivity measurements define an important dispersion due to the low conductivity of such material. The measurements of conductivity LLDPE + GTR in the zone corresponding to the high frequencies of the linear dependence of the conductivity with the frequency are observed in the zone corresponding to the high frequencies of the log-log plots. The conductivity dependency on the frequency is seen as linear. This fits a sublinear dispersive conductivity model Equation (1), as it is common in polyethylene and similar materials [46], (Equation (1)):(1)$$\sigma =\sigma_{0}+A\omega^{n}$$
where σ_0_ is the conductivity to direct current (DC), ω=2πf, where f is the frequency, while A and n (which have values between 0 and 1) are parameters that depend on temperature and materials. This equation implies two different behaviors, one at low frequencies where DC term is dominant and there is no dependency between conductivity and frequency, and another dispersive in which the conductivity has a potential dependence with the frequency. As ω increases, the dispersive behavior appears and replaces the DC one. Therefore, in Figure 4a, for low frequencies, the conductivity is seen in a direct current (DC) regime, and, for high frequencies and all the composites, the AC current regime dominates. Figure 4a shows the conductivity for a wide range of frequencies (0.01 Hz–3 MHz) and combines two differentiated conductivity regimes: DC (low frequencies) and AC (high frequencies), in a low-frequency regime, the conductivity measurement domains in DC conductivity regime with the addition of higher GTR loads, but, for the rest of the composites, the AC conductivity regime prevails at 50 Hz.

In Figure 4a,b, it is observed that the neat LLDPE has a very low DC conductivity. For most temperatures and GTR concentrations, the frequency of trend change that delimits one or the other behavior is below the range of frequencies analyzed. As is seen in the Figure 4a, only at high GTR concentrations is it possible to clearly see the change in the slope in the low-frequency region of the spectrum (10^−2^ and 100 Hz for LLDPE + 70%GTR and LLDPE + 50%GTR, respectively). The reason is that, by increasing the concentration of GTR, and therefore the amount of carbon black (CB) in the sample, the DC conductivity increases and shifts the rate of behavior change to higher values. As expected, there is also a direct relationship between GTR concentration and conductivity, both in direct current and the dispersion behavior. The CB present within the GTR particles, nearly 35% as the TGA test in GTR samples [47], is much more conductive than insulation polymers and is generally used to improve the electrical properties of these materials [48,49]. However, the increase in conductivity is not enough to cause the material to lose its insulation condition. 

On the other hand, in Figure 4b, it is observed how the conductivity at low frequencies (50 Hz) is greatly increased when measured at different temperatures (30 to 100 °C), and, for different GTR percents in the polymer blends, the samples increase nearly 4 orders of magnitude, when the percent of GTR is increased (from 0% to 70% of GTR). For the increase in temperature, the changes in conductivity with temperature are not significant, so they are not important changes in the slope by temperature, but the changes with the presence CB are significant. So, from the observation of Figure 4a,b, we can obtain the next conclusions, since LLDPE has a very low DC conductivity. For GTR concentrations, the crossover frequency that delimits one or other behaviors is located below the range of frequencies analyzed. Only at low frequencies for high GTR concentrations is it possible to see clearly the change in the slope at the low-frequencies region of the spectra analyzed in Figure 4a. The reason is that by increasing the frequency and the GTR concentration, and therefore the CB amount in the composite material, the DC conductivity rises and shifts the crossover frequency to higher values. 

The behavior in Figure 4b and Figure 6 (frequency fixed at 50 Hz) is coherent with electrical insulators; temperature does not affect the conductivity process in the range of temperature (30 to 100 °C). The conductive mechanism observed for the temperature range is that the valence band is full and energetically separated from conduction, and the barrier between them impedes the electrons be excited for charge transport. So, the temperature is not high enough, for electrons to move over the barrier. The major charge carriers are the electrons and the hole in the conduction band and valence band, respectively. The basic parameters to these are thought to be the concentrations and mobility (µ) of electrons and holes, which measure the movement capacity of charge carriers in the composite [50,51]. The hopping mechanism transport charge is suitable for polymeric materials [52]; this process is performed when two different charge carriers, such as electrons and holes, are separated by a potential barrier, one can move to the other through tunneling the barrier or moving over the barrier via an activated state [53].

### 3.3. Permittivity

Figure 5 shows, for different compounds of LLDPE/GTR, the values of real permittivity (ε′) and imaginary permittivity or dielectric loss factor (ε″), which are proportional to the stored and dissipated energy, respectively, in each cycle, in relation to the frequency, at a temperature of 30 °C. At low frequencies, both real permittivity (ε′) and dielectric loss factor (ε″) increase as GTR content rises. The real and imaginary permittivity decreases as frequency increases for the compounds with the greatest presence of GTR (50% and 70%LLDPE/GTR compounds), as this tendency is less significant for lower GTR concentrations (LLDPE + 5%GTR – LLDPE + 10%GTR – LLDPE + 20%GTR). In the case of LLDPE without GTR, ε′ does not depend on the frequency. The drops in the dielectric dispersion [54,55] contribute to such phenomenon only because of GTR contents (and the majority CB). Because LLDPE is a non-polar polymer, how can you verify in the LLDPE pure or low GTR contents composites (Figure 5a,b)? Similar decreases are observed in all the samples studied for the ε″ results. In this case, there are contributions of the conductance ($\varepsilon \prime \prime \propto \frac{\sigma }{\varepsilon_{0}\omega }$) and interfacial phenomena at low frequencies.

The analysis of the permittivity (Figure 5a,b) brings a clear distinction in the behavior of a Debye system from the Maxwell–Wagner–Sillars model (MWS) system. The frequency response for the LLDPE scale superlattices can be directly modeled by MWS (see Figure 5a,b) [56], more than a Debye expression, which cannot account for the low-frequency behavior. Such a low-frequency relaxation suggests that the ions can move over long distances and may affect the permittivity [57]. The explanation for this phenomenon is that in heterogeneous systems, where we find two or more components or phases, such as LLDPE + GTR, the dielectric relaxation of the MWS type occurs. This phenomenon has a critical point or maximum at 50%GTR reinforcement particles, since at this point the composite is in a state of maximum heterogeneity, due to its own composition. So, at 50%GTR, the interfacial polarization due to the MWS phenomenon is maximum. 

On the other hand, real permittivity (ε′) and dielectric losses (ε″) at a frequency of 50 Hz and at various temperatures between 30 and 100 °C are shown in Figure 6a,b. Higher GTR contents are directly related to higher ε′ and ε″ values. There is 1 order of magnitude between the ε′ of pure low-density linear polyethylene and the 70% GTR samples analyzed, and 4 orders of magnitude between the ε″ of pure LLDPE and the LLDPE + 70%GTR samples analyzed. In general, the permittivity seems to maintain when the temperature rises for the real permittivity (ε′) and for imaginary permittivity or the dielectric loss factor (ε″). For the LLDPE/GTR compounds, the values are slightly increasing with the increase in the temperature. The data obtained from the ε″ pure LLDPE sample show a relaxation with a peak around 70–80 °C, and then begin to increase again. Since the GTR is more polar and conductive than low-density linear polyethylene, the dielectric relaxation of the polymer matrix is masked by the filling properties (GTR). 

The permittivity (ε′) and dielectric loss (ε″) for a frequency of 50 Hz and several temperatures between 30 and 100 °C are shown in Figure 6. The GTR higher contents of LLDPE compounds are directly related to higher ε′ and ε″ values. There is one order of magnitude between the real permittivity (ε′) of the LLDPE and the 70%GTR/LLDPE samples, and there are about 1.5 orders of magnitude in case of the dielectric loss factor (ε″). Temperature does not affect the ε′ behavior for the compounds analyzed (LLDPE/GTR), and it affects very mildly the ε″ behavior, only nearly the 100 °C.

In Figure 7, the LLDPE + 20% GTR compound is analyzed, in the range of frequencies (from 10^−2^ to 3 × 10^6^ Hz) and for a range of temperatures, from 30 to 100 °C. From this analysis, the conclusion is clear. For ε′, there is a peak in frequency from 10^−1^ to 100 Hz and which accentuates with increasing temperature and shifts it to higher frequencies, and ε″ clearly tested that Maxwell–Wagner–Sillars (MWS) relaxation is clearly seen and that the type interfacial polarization processes acts from 10^2^ Hz. It is also checked that, in this frequency value, the behavior of ε″ no longer depends on the temperature, only on the frequency. This process is clearly seen in Figure 7b, where we see the only dependence on the ε″ with the frequency from 10^2^ Hz to higher levels of frequency. It is certainly explained by MWS interfacial polarization processes [58,59,60,61].

### 3.4. Dielectric Modulus (M″)

Different authors [62,63] have studied the electrical module that relates the process of dielectric relaxation, the polarization of the interface or the Maxwell–Wagner–Sillars effect. In polymeric compounds, the relaxation phenomena in the low-frequency region are attributed to the heterogeneity of the systems. The investigation of these electrical processes is carried out through the electrical module. The Debye model, Cole–Cole, Davidson–Cole and Havriliak–Negami equations for dielectric relaxation are expressed in the form of an electrical module. Relaxation interfacial phenomena in heterogeneous materials are usually located at very low frequencies so they are not visible in low-temperature measurements for the frequency range studied. As they are thermally activated processes, they would appear in this frequency range for higher temperatures. However, at such frequencies and temperatures, there are several phenomena that typically darken the interfacial relaxations (electrode polarization, conduction phenomena). In order to overcome these problems, it can be convenient to use the formalism of the Electric Modulus [64] (see Equation (2)). At low-frequency behavior, the analysis of the imaginary modulus (M″) provides us the features of the dielectric loss factor avoiding any contribution of the conductance. In Figure 8, one can see how the relaxation peak is going from higher frequencies (3·10^6^ Hz) for the compounds with a low presence of GTR in the compounds blends (from 0% to 10% GTR compounds), to lower frequencies (100 Hz) for the compounds with higher contents of GTR (50% and 70% of GTR), so the presence of GTR is affecting the relaxation phenomena of LLDPE and changing the frequency relaxation. This trend is clearly shown in Figure 8.
(2)$$M=\frac{1}{\varepsilon }=\frac{1}{\varepsilon^{\prime }-j\varepsilon^{\prime \prime }}=\frac{\varepsilon^{\prime }}{{\varepsilon^{\prime }}^{2}+{\varepsilon^{\prime \prime }}^{2}}+j\frac{\varepsilon^{\prime \prime }}{{\varepsilon^{\prime }}^{2}+{\varepsilon^{\prime \prime }}^{2}}=M^{\prime }+jM^{\prime \prime }$$

The modulus analysis (see Figure 8) is used for analyzing and determining the dynamical aspects of electrical transport phenomena. The analysis gives the correlation between the motions of mobile charge [65]. Figure 8 shows the variation of imaginary modulus (M″) with frequency at different GTR loadings. In the accessible frequency range, the spectrum exhibits one relaxation peak for each GTR loading. The peaks shift systematically toward lower frequencies with an increase in GTR loading. The broadening of the peak indicates the spread of relaxation time with different (mean) time constants, and hence a non-Debye type of relaxation in the materials is observed that is coherent with the Maxwell–Wagner–Sillars model (MWS) system, and this is consistent with the permittivity data. The nature of the processes is further explored using the Argand Diagram (complex plane representation) for dielectric modulus, as shown in Figure 11, for different concentrations of GTR loading. The curve of LLDPE + 50%GTR presents a peak at a slightly lower frequency than LLDPE + 70%GTR (see Figure 8), which means a better charge mobility for this composite at low frequencies [66]. This behavior is coherent with the Maxwell–Wagner–Sillars model (MWS) system.

Figure 9 simultaneously shows the evolution of the imaginary module (M″) with temperature and frequency for the pure LLDPE and six different concentrations of LLDPE and GTR (5%, 10%, 20%, 30%, 50% and 70% GTR). At low frequencies and high temperatures, a relaxation associated with the presence of GTR can be detected, like we have shown in the last Figure (Figure 8). Firstly, this relaxation appears as a smooth hump on the slope of the α’ relaxation. By increasing the GTR amount in the compound, this peak becomes more prominent. Such high temperatures and low-frequency-GTR-associated relaxation can be identified with interfacial polarization phenomena, that is, a Maxwell–Wagner–Sillars relaxation [67]. These phenomena are typical of heterogeneous materials in which there are regions with different conductivities. The results in Figure 9 reveal that, for low GTR concentrations, the α′ peak is placed at very low frequencies. However, it seems that, for GTR concentrations of 5%, the relaxation is greater than of pure LLDPE. For higher GTR concentrations, the α′ relaxation decreases, shifts to higher frequencies, and, finally, it almost vanishes for 70%. Such evolution is analogous to that of crystallization, which is higher for 5% of GTR and drops if the concentration increases. The fact that α′ relaxation should be related to amorphous-crystal interfaces is consistent with this apparent direct dependency.

### 3.5. LLDPE/GTR Activation Energies (Ea)

The DC conductivity follows the Arrhenius-type thermal activation process [68,69]:(3)$$\sigma_{dc}=\sigma_{0}\cdot e^{\left(-\frac{E_{a}}{kB\;\cdot \;T}\right)}$$
where $\sigma_{0}$ is a constant, Ea is the activation energy and kB is the Boltzmann constant. The activation energy has been calculated from the linear fitting of log ($\sigma_{dc}$) versus 1/T (Figure 10), and the results are shown in the Table 2. Above 70–100 °C, the activation energy of each composite has low variation when changing temperature. Since the conductivity of the carbon black present in the GTR is much higher than LLDPE, the decrease in log ($\sigma_{dc}$) can be explained by assuming that the space charge mobility rises with GTR. Differences in the activation energy above 100 °C can be explained by the differences of the conductivity in each composite. Despite being the evolution in the Ea parameter, the changes observed are not very important. Another interesting issue that is worthy of comment is the presence of only one lineal region for the plot of dc conductivity against 1/T. This means that in the interval of temperature remain only crystalline structures (LLDPE Tm is 122 °C). As depicted in Figure 10, at temperature below Tm, the DC conductivity has been observed to decrease gradually with the temperature up to 70 °C. This is due to the fact that, at high temperatures (i.e., 100 °C), the energy would be large enough to increase the free volume in the system, which facilitates the motion of small GTR particles. When the amount of GTR in the compound increase (i.e., 40, 50%), the Ea increase due to the possibility of producing GTR clusters that reduce the segmental motions of big particles of GTR. Although the addition of GTR, which contains carbon black, increases the DC conductivity, this reduction in motion possibilities avoids a significant increase in DC conductivity.

### 3.6. LLDPE/GTR Argand Diagram

Representation of the impedance in a vector diagram or Argand diagram, using the Module Electric, that is, a complex magnitude: *M** = *M′* + *j**M″*(4)

Coelho’s theory of space charge contribution [70] is based on the concept of macroscopic dipole. A sample with electrical neutrality, has the mobile carriers evenly distributed in the absence of field. If an electric field is applied to the sample, the mobile carriers move towards the opposite sign electrode, leaving opposite sign carriers next to the other electrode, resulting in equilibrium after the next charge distribution. This distribution constitutes a macro dipole that would oscillate in an alternating field with the frequency of the field, causing a relaxation process that affects the heat of the permittivity of the medium. For the development of the model, it is assumed that the processes that regulate electrical charge transport are ohmic conduction and diffusion caused by the concentration gradient of the carriers. Coelho has shown that the Argand diagram corresponding to the relaxation of space loading is very similar to the Cole–Cole circle [71]. In Figure 11, M″ is represented related to M′ in the so-called Argand’s plot for the electric modulus at 120 °C. An arc at the low-frequency behavior is observed for all the samples. This is consistent with the Coelho’s model for space charge relaxation. The relaxation of Coelho is observed for pure LLDPE in the Figure 11. In the rest of the analyzed cases of changes for different GTR contents in LLDPE + GTR blends analyzed, changes in the Argand Diagram are seen as a larger percentage of GTR is incorporated in the LLDPE blends. Coelho’s Model assumes that, when an electric field is applied to a sample, the free charges move through the sample toward the electrode of opposite sign and finally the accumulation of charges close to electrodes results in a macro dipole [72,73]. If the field oscillates, then the macro dipole is forced to oscillate with the frequency of the field, and a dipolar-like relaxation appears. A semicircular trend (a requirement for non-Debye model) followed by a linear increase is observed for all GTR loadings in the Argand Diagram (Figure 11). It is also observed that, with the increase in GTR loading, the size of the semicircular loop decreases, which confirms better conduction. So, the presence of a non-Debye type of relaxations has been confirmed by complex dielectric modulus analysis [74]. It can also be observed that the arc is an almost perfect semicircle for pure LLDPE, which implies that the electrodes are blocked, and can conclude that a macro-dipole behavior for neat LLDPE, as is tested in the Figure 11.

### 3.7. Industrial Applications Analysis

The addition of increasing amounts of GTR in the LLDPE matrix has worsened the electrical insulation properties of the composites but it is seen that there are composites that, for low contents of GTR (5%, 10% and 20%), have enough insulation properties according with UNE Standards (Spanish Standard) and International Standards (IEC), as we can check in Table 3 and Table 4. This give possibilities to use composites with low amounts of GTR for low-requirement applications, such as: universal electrical cable joint, filler for electrical applications, pipes for electrical cables, as you can see in Table 4. As is seen in Table 3, electrical parameters, needed for insulators, such as conductivity, loss factor and loss tangent, are slightly affected by GTR loads, for low GTR amounts (5%, 10%, 20% of GTR), but the worsening of these electrical properties (Table 3) is significant for higher amounts of GTR particles in the composites (40%,50%,70% of GTR). Loss Tangent, Tg δ = ԑ″/ԑ′.

UNE and IEC standards are used in order to compare values of specific industrial electrical applications with dielectric values [75], obtained from experimentation about mixtures of different LLDPE + GTR particles. Different electrical applications, such as those selected in Table 4, have been analyzed. However, it has been found that the analyzed mixtures for low GTR (5%–10%–20%) and LLDPE + GTR amounts produce materials that are insulating enough to constitute electrical insulators for different applications according to Table 4 (conductivity <10^−12^ S/cm and Tangent of δ <10^4^), since, in the results analyzed in Table 3, all the samples analyzed with 5%–20% GTR have turned out results acceptable, and within limits, as established by analyzed UNE and IEC standards. As a conclusion, 20% GTR analyzed composites have acceptable levels of electrical insulation, for the following selected applications: universal electrical cable joint, filler for electrical applications, cable tray systems, cable ladder system and pipes for electrical cables.

## 4. Conclusions

Conductivity (σ) as well as real permittivity (ԑ′) and dielectric loss factor (ԑ″), in the LLDPE blended with GTR particles, increase with the GTR concentration of the analyzed compounds (LLDPE + GTR). As it takes place in similar materials [76,77], the conductivity presents a sublinear dispersive behavior in the compounds until 40%GTR. The increase in conductivity is higher in the low-frequency behavior (for 70%GTR/LLDPE, it is around 10^4^ times higher than for LLDPE), but, in any case, the samples relatively maintain their insulating properties. On the other hand, permittivity decreases with frequency for higher GTR concentrations (50–70%GTR), whereas the LLDPE/0–20%GTR real permittivity does not depend on the frequency. The analysis with the temperature shows that the real permittivity slightly decreases whilst the dielectric loss factor increases with temperature, except for pure LLDPE, which presents its own dielectric complex spectrum. The dielectric modulus formalism has enabled us to study the MWS relaxation due to interfacial polarization. For GTR concentrations lower than 20%, this relaxation is not significant. The activation energy shows a big difference between the LLDPE net and LLDPE + GTR composites (Table 1) due to the increase in the present of carbon black in the percent’s of GTR. From the Argand’s plot [78], is proved that electrodes are almost blocked in the pure LLDPE sample and they become less blocked when increasing the GTR content showing a macro-dipole behavior for LLDPE sample. Finally, although the insulating features are reduced with GTR addition, especially above 20% GTR, the conditions for insulation applications are maintained for low-load amounts of GTR composites (5%–10%–20%), and the composites can be adopted in industrial applications due to the samples of LLDPE/GTR maintaining insulation properties as we have proved previously. The results obtained from the analysis of these compounds show that, at 20% GTR, which corresponds to the maximum degree of dispersion of GTR particles, is the maximum amount of value for keeping acceptable values of the conductivity of the compound. This would allow its use in various fields of industry, a priori, as an insulator with low requirements, such as a universal electrical cable joint, filler for electrical applications and cable tray systems and cable ladder system and pipes for electrical cables.

## Figures and Tables

**Figure 1 polymers-12-01075-f001:**
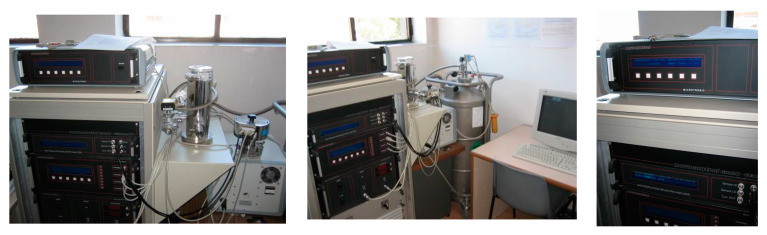
Dynamic Electric Analysis (DEA) equipment used with the control unit for the parameters of the test.

**Figure 2 polymers-12-01075-f002:**
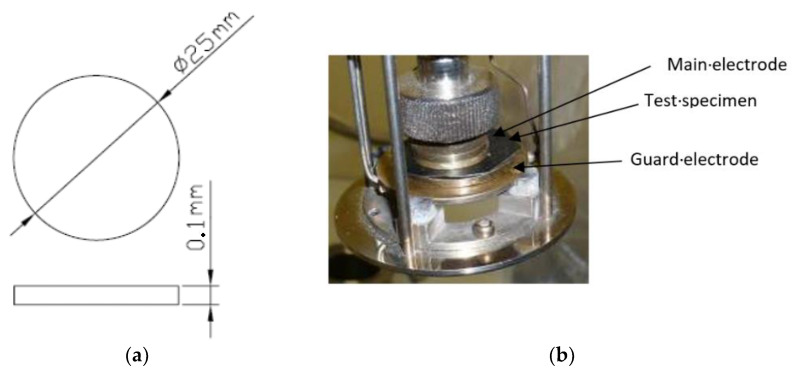
Samples for Dynamic Electric Analysis (DEA) used. (**a**) Specimen dimensions; (**b**) DEA electrodes with composite sample.

**Figure 3 polymers-12-01075-f003:**
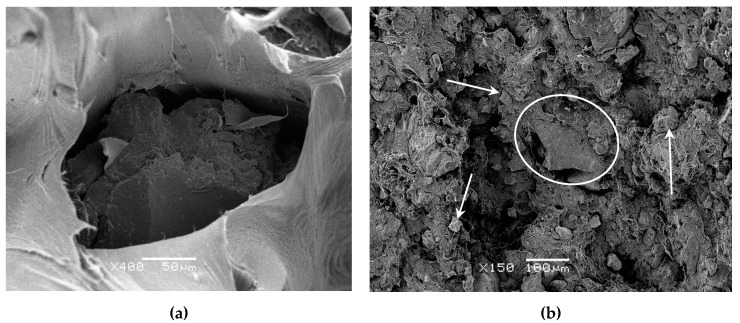
Scanning Electron Microscopy (SEM) micrographs of LLDPE/20%GTR compounds: (**a**) a magnification of 400 and (**b**) a magnification of 150 that shows the dispersion of small particles (arrows) and lack of compatibility large particles (circle).

**Figure 4 polymers-12-01075-f004:**
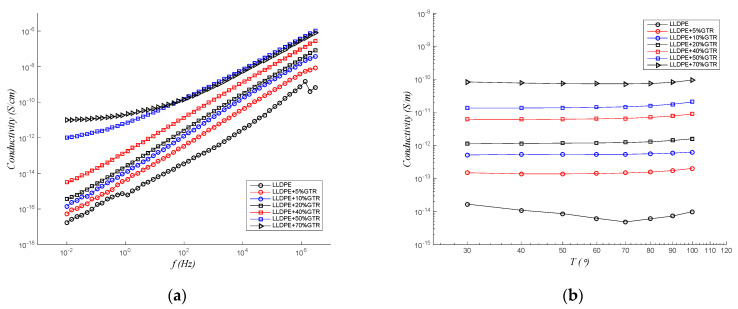
Frequency and thermal response of the conductivity of LLDPE/GTR compounds (**a**) at 30 °C with a frequency range of 10^−2^ to 3 × 10^6^ Hz and (**b**) at 50 Hz with a temperature range of 30 to 100 °C.

**Figure 5 polymers-12-01075-f005:**
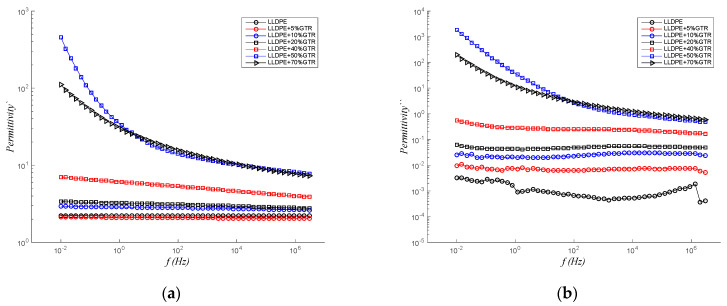
Compounds LLDPE/GTR, at 30 °C in relation to frequency (From 10^−2^ to 3 × 10^6^ Hz). (**a**) Real permittivity, ε′, and (**b**) dielectric losses factor, ε″.

**Figure 6 polymers-12-01075-f006:**
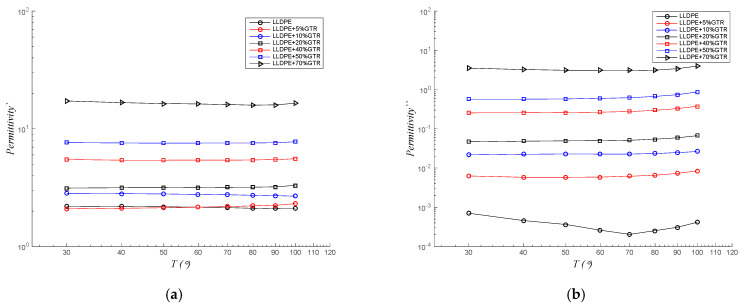
LLDPE/GTR compounds at 50 Hz in relation to temperature (from 30 to 100 °C); (**a**) real permittivity, ε′; (**b**) ε″, dielectric losses factor.

**Figure 7 polymers-12-01075-f007:**
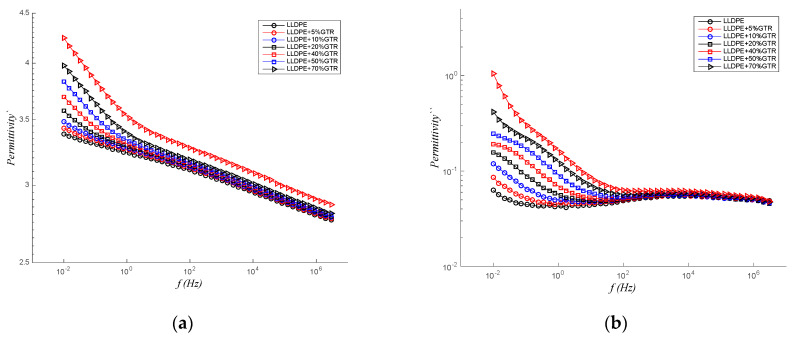
Real permittivity (**a**) and imaginary permittivity or dielectric loss factor (**b**) at various temperatures for the LLDPE/GTR-20% compound analyzed.

**Figure 8 polymers-12-01075-f008:**
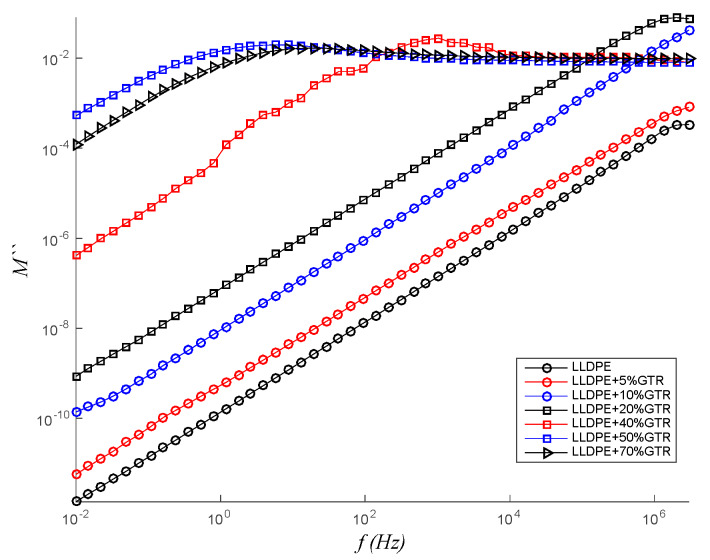
Imaginary electrical module (M″) of the LLDPE/GTR at 120 °C depending on the frequency, which ranges from 10^−2^ to 3 × 10^6^ Hz.

**Figure 9 polymers-12-01075-f009:**
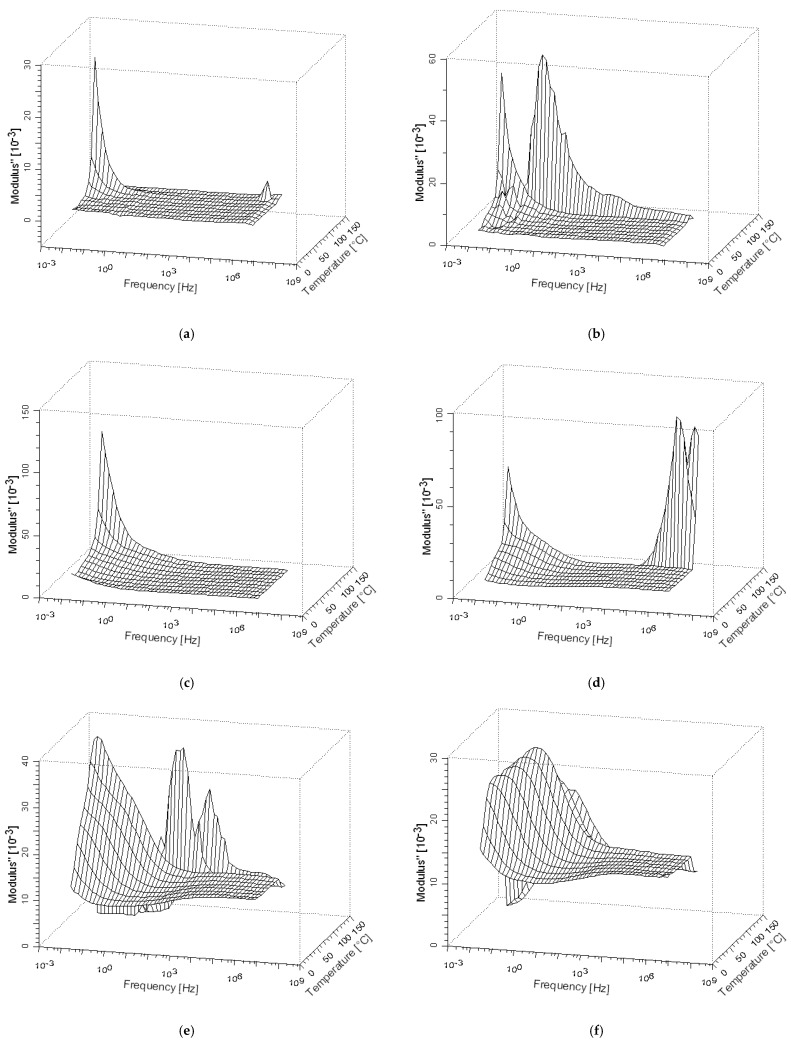
Three-dimensional diagrams of the imaginary component of the electric module (M″) in relation to temperature and frequency for LLDPE (**a**), LLDPE/GTR-5% (**b**), LLDPE/GTR-10% (**c**), LLDPE/GTR-20% (**d**), LLDPE/GTR-40% (**e**), LLDPE/GTR-50% (**f**), LLDPE/GTR-70% (**g**).

**Figure 10 polymers-12-01075-f010:**
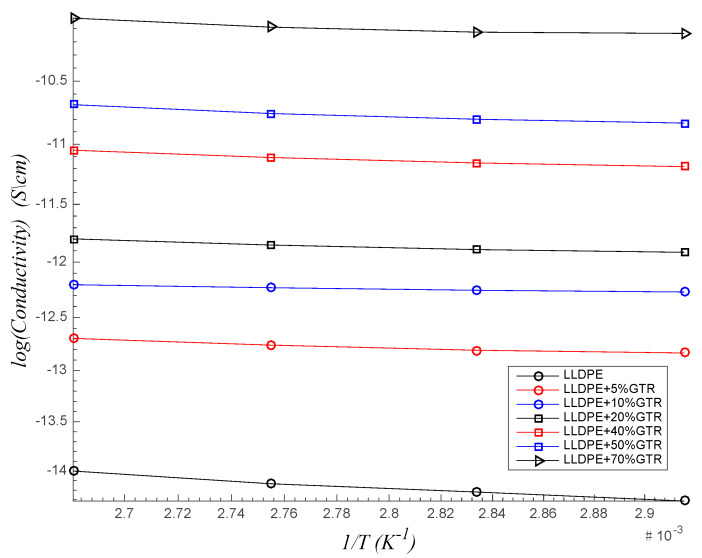
Linear fitting of log (σ_dc_) versus 1/T for composites with different amounts of GTR.

**Figure 11 polymers-12-01075-f011:**
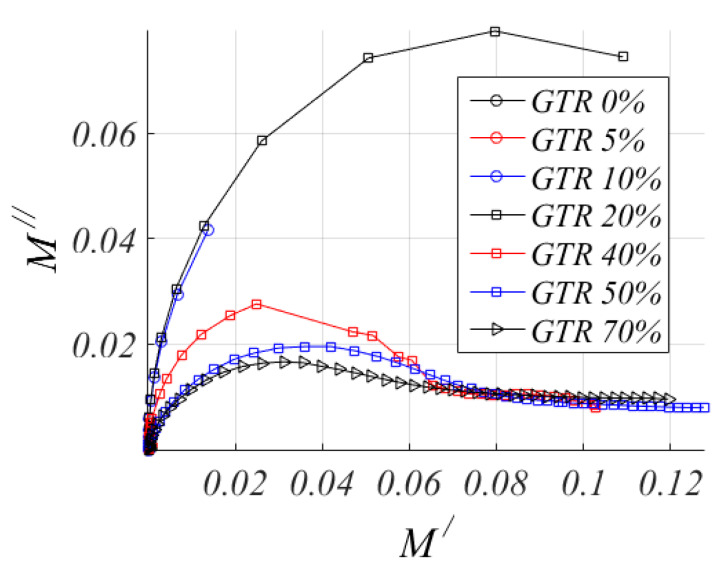
Argand Diagram (M″-M′) for LLDPE/GTR blends at a temperature of 120 °C.

**Table 1 polymers-12-01075-t001:** Linear low-density polyethylene (LLDPE) features provided by the manufacturer and processing variables for composites (LLDPE-GTR) with the Brabender mixing machine.

Properties	Unities	Values
Flow index	g/10 min	1.0
Density	g/cm^3^	0.924
Melting temperature	°C	126
Tensile strength	Psi	8.000
Elongation at break	%	720
Processing Temperature	°C	124 °C
Pressing Temperature	°C	130 °C

**Table 2 polymers-12-01075-t002:** Activation energy and conductivity rise in the range of temperatures from 70 to 100 °C.

Ea (eV)	$∆\sigma =\sigma_{100\;{}^{\circ}C}-\sigma_{70\;{}^{\circ}C}\;(S/m)$	GTR (%)
0.2585	0.0001 × 10^−10^	0%
0.1157	0.0005 × 10^−10^	5%
0.0553	0.0009 × 10^−10^	10%
0.0982	0.0038 × 10^−10^	20%
0.1130	0.0238 × 10^−10^	40%
0.1220	0.0590 × 10^−10^	50%
0.0959	0.2200 × 10^−10^	70%

**Table 3 polymers-12-01075-t003:** Conductivity, loss factor and loss tangent, for each composite, at 50 Hz and 30 °C.

Composite	Conductivity (S/cm)	Loss Factor (ԑ″)	Loss Tangent (Tg δ)
LLDPE	1.6739 × 10^−14^	7.0115 × 10^−4^	3.19 × 10^−4^
LLDPE + 5% GTR	1.4927 × 10^−13^	6.2523 × 10^−3^	1.49 × 10^−3^
LLDPE + 10% GTR	5.1646 × 10^−13^	2.1633 × 10^−2^	7.67 × 10^−3^
LLDPE + 20% GTR	1.1145 × 10^−12^	4.6681 × 10^−2^	1.49 × 10^−2^
LLDPE + 40% GTR	6.0551 × 10^−12^	2.5363 × 10^−1^	4.64 × 10^−2^
LLDPE + 50% GTR	1.3597 × 10^−11^	5.6954 × 10^−1^	7.47 × 10^−2^
LLDPE + 70% GTR	8.3718 × 10^−11^	3.5067 × 10^0^	2.04 × 10^−1^

**Table 4 polymers-12-01075-t004:** Applications analyzed according to different standards, and the electrical criterion for them.

Applications Analyzed	Electrical Criterion	Standard
1. Universal electrical cable joint	Conductivity <10^−12^ S/cm	UNE-HD 628; IEC 60840
2. Filler for electrical applications	Conductivity <10^−12^ S/cm	UNE-HD 632; UNE-EN 60811-4-1; UNE-ISO 1853:2012
3. Cable management—Cable tray systems and cable ladder system. Pipes for electrical cables	Conductivity <10^−12^ S/cm	UNE-EN 61537:2007; UNE-EN 50085-1, IEC 61537:2006

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
