# Peer review of "Study and Characterization of the Dielectric Behavior of Low Linear Density Polyethylene Composites Mixed with Ground Tire Rubber Particles"

_polymers, 2020, doi:10.3390/polym12051075_

Round 1

Reviewer 1 Report

This manuscript studies the dielectric properties of low linear density polyethylene (LLDPE)/ ground tire rubber (GTR) composites, which sheds light on the re-utilization of GTR in industrial applications. However, as I examine the details about the dielectric parameters of the prepared composite materials with increased electrical conductivity, loss tangent as well as largely decreased conduction activation energy, which seems that thus designed composite is not a good candidate to be used as an insulation material. I, therefore, do not recommend the publication of this paper. Other issues I have as listed below.

  1. There is no necessary to use the abbreviations of LLDPE and GTR in the title
  2. Abstract: Line 27. Is the DEA test corresponds to the “Dynamic Electrical Test”?
  3. Too many keywords
  4. The introduction part is not well written. The authors are suggested to summarize the recent progress in recyclable polymer materials for industrial applications. Additionally, it is no doubt that the carbon black-containing polymer composites can be used for electrostatic dissipation and electromagnetic interference shielding, but it is not suitable for the electrical insulation application.
  5. Figures 3a and b show very incompact interfaces between LLDPE and GTR phases, which may significantly reduce the mechanical strength of the resulting composites and restrict the engineering application.
  6. There lacks analysis in Section 3.4 and 3.6.

Author Response

Response to Reviewer 1 Comments

Point 1: This manuscript studies the dielectric properties of low linear density polyethylene (LLDPE)/ ground tire rubber (GTR) composites, which sheds light on the re-utilization of GTR in industrial applications. However, as I examine the details about the dielectric parameters of the prepared composite materials with increased electrical conductivity, loss tangent as well as largely decreased conduction activation energy, which seems that thus designed composite is not a good candidate to be used as an insulation material. I, therefore, do not recommend the publication of this paper. Other issues I have as listed below.

Response 1: Thank you for your comments. First of all, as you introduce in your comments, we have checked that the addition of increasing amounts of GTR in the LLDPE matrix, as a result in the present research we can see that there are composites that for low contents of GTR (5%, 10% and 20%) have enough insulation properties according with UNE Standards and IEC Standards (Spanish Standard and International Standards, respectively), as we can check in new introduced tables 3 and 4, this give possibilities to use composites with low percent’s of GTR for a low requirements applications such as: universal electrical cable joint, filler for electrical applications, cables tray systems, and pipes for electrical cables, as you can see in a new table 3 incorporated in paper.

New tables 3 and 4 are incorporated in paper, in a new section 3.7 Industrial applications analysis: Table 3. Conductivity, loss factor and loss tangent, for each composite, at 50 Hz and 30ºC and Table 4. Applications analyzed according with different standards, and electrical criterion for them. And a development about composites industrial applications, has been introduced in a new section of article 3.7 Industrial applications analysis.

Limits of electrical conductivity for applications: UNE contributes to the development of the economy through the elaboration of technical standards and international cooperation activities. UNE is the standardization agency in Spain, it promotes the development of quality infrastructure, promoting the transfer of knowledge and the strengthening of companies, and are used for validation composite applications [1]. On the other hand, the IEC (International Electrotechnical Commission) is the world’s leading organization for the preparation and publication of International Standards for all electrical, electronic and related technologies. With this two Standards, UNE and IEC, are used in order to compare values of specific industrial electrical applications.

We have discarded the electrical cables application, because the high and big quantity requirements for this application, make it not suitable. On the other hand, we have searched low requirements applications such as the selected in table 4. Electrical insulating characteristics of the composites LLDPE+GTR worsen with the addition of GTR and therefore it produces insulating mixtures of worse characteristics, with the incorporation of GTR, however it has been found that the analyzed mixtures behave, in general, as good electrical insulators, according to the analyzed standards (Table 3), for low contents of GTR percent’s (5-10-20%), LLDPE+GTR produce materials that are insulating enough to constitute electrical insulators for different applications (Conductivity <10-12 S/cm and Tangent of δ <104) since the results analyzed in table 3 in all the samples analyzed with 5-20% GTR has given results well below the limits, as established by analyzed UNE and IEC standards.  We have introduced the table and the comments in the section 3.7. This study allows to answer the next question: For what percentage of GTR, in a LLDPE polymer composites, present electrical properties acceptable regards to industrial applications as electrical insulators? We can answer to this question from the previous table that until 20% GTR the properties have acceptable levels of electrical insulation.

Reference:

[1] Marín-Genescà, M.; García-Amorós, J.; Mujal-Rosas, R.; Salueña Berna, X.; Massagués Vidal, L. Comparison of Mechanical and Electrical Characteristics of Various Polymers Blended with Ground Tire Rubber (GTR) and Applications. Appl. Sci. 2019, 9, 1564.

Point 2: There is no necessary to use the abbreviations of LLDPE and GTR in the title

Response 2: Thank you for this comment, we are completely agreed with this appreciation, we have changed the title of the article: “Study and Characterization of Dielectric Behavior of Low Linear Density Polyethylene Composites Mixed with Ground Tire Rubber Particles”. Is more suitable the new title without the abbreviations GTR and LLDPE.

Point 3: Abstract: Line 27. Is the DEA test corresponds to the “Dynamic Electrical Test”?

Response 3: Thank you for your question. No, the analysis performed is Dynamic Electric Analysis or, also called Dielectric Analysis, DEA, as have developed in some different research works and reviews [2]. DEA is used to study of dielectric materials behavior by means of time-varying electric fields allows to obtain information about the material polarization mechanisms. Thereby, the physical parameters of any relaxation can be obtained. To characterize a dielectric material, several time functions for the applied electric field can be used. For instance, the step function is a sudden change in the electric field that leads to the so-called Time-Domain Response. Another case consists in applying a sinusoidal field with a determined frequency. Thus, the Frequency-Domain in DEA, the material is exposed to an alternating electric field generated by an applied sinusoidal voltage.  In this technique, a variable electric field is applied to the sample and its response is recorded. With the experimental DEA equipment, materials dielectric behavior can be studied in a wide frequency range and at different temperatures, in our research from 1·10-2 Hz to 3·106 Hz and from 30ºC to 120ºC respectively. Samples are submitted a dielectric to electrical fields, the dependence with frequency of both permittivity (ε (ω)) and conductivity (σ (ω)) can be determined. The applied electric field causes the alignment or induction of dipoles in the material which results in polarization. One feature of DEA is that this spectroscopy allows for the investigation of molecular mobility, or relaxations of the material response is obtained.
Reference:
[2] Brent Hilker Kimberly B.FieldsAbrahamSternBrianSpaceX. PeterZhangJulie P.Harmon. Dielectric analysis of poly(methyl methacrylate) zinc(II) mono-pinacolborane diphenylporphyrin composites. Volume 51, Issue 21, 1 October 2010, Pages 4790-4805. Polymer. https://doi.org/10.1016/j.polymer.2010.08.049

Point 4: Too many keywords

Response 4: Completely agreed with this comment, we have reduced the Keywords: Reused tires; LLDPE; Electrical properties; Electrical modulus; Composite recycling applications. Erasing Conductivity and Permittivity, because they content in the Electrical properties

Point 5: The introduction part is not well written. The authors are suggested to summarize the recent progress in recyclable polymer materials for industrial applications. Additionally, it is no doubt that the carbon black-containing polymer composites can be used for electrostatic dissipation and electromagnetic interference shielding, but it is not suitable for the electrical insulation application.

Response 5: Agreed with your comment, we can see the incorporation of mentions to the electrostatic applications is not suitable for the Introduction, for this we have erased this references and mentions to electrostatic, is not the aim of the article, so we have changed the introduction erasing this mentions. On the other hand, as is suggesting the reviewer, we have introduced a summary about the recent progress in recyclable polymer materials for industrial applications, as you can see in the resubmitted article version.

Point 6: Figures 3a and b show very incompact interfaces between LLDPE and GTR phases, which may significantly reduce the mechanical strength of the resulting composites and restrict the engineering application.

Response 6: As reviewer remarked, we can see this phenomena, very incompact interfaces between LLDPE and GTR phases, we are agree with this appreciation, however, in this paper we have not presented the results obtained in the mechanical essays performed, and we want to present them, deeply in a upcoming paper, however, we may also see how for small percentages of GTR (5-20%) we can obtain acceptable mechanical properties, as we can check in similar experiences [3,4]..

References:

[3] Mujal, R.; Marin, M.; Orrit, J.; Rahhali, A.; Colom, X. Dielectric, mechanical, and thermal characterization of high-density polyethylene composites with ground tire rubber. J. Thermoplast. Compos. Mater. 2012, 25, 537–559.

[4] R. Mujal Rosas, M. Marin Genesca, J. Garcia Amoros, Xavier Salueña Berna, Xavier Colom Fajula. Influence on the mechanical properties of various polymeric composites reinforced with GTR particles. Afinidad. ISSN 0001-9704, Vol. 76, Nº. 588, 2019, págs. 241-25

Point 7: There lacks analysis in Section 3.4 and 3.6.

Response 7: Thank you for your comments, we have introduced new analysis in the sections 3.4 and 3.6, as requested the reviewer. An analysis text was developed for inclusion in the article and is as you can see in the resubmitted article version, in 3.4 and 3.6 sections: 3.4. Dielectric Modulus (M’’); 3.6. LLDPE/GTR Argand Diagram. Analysis about non Debbie dielectric behavior are provided, in this section [5].

Reference:

[5] S. K. Tiwari, R. N. P. Choudhary and S. P. Mahapatra. Dynamic mechanical and dielectric relaxation studies of chlorobutyl elastomer nanocomposites: Effect of nanographite loading and temperature. High Performance Polymers 2015, Vol. 27(3) 274–287. DOI: 10.1177/0954008314545137

Reviewer 2 Report

The submitted paper reports on the dielectric properties of polymer blends made of low-density polyethylene mixed with various amounts of reused tire powders. The overall goal of the paper is to find possible applications of reused tires. This goal is very appealing and the paper, if properly revised and re-reviewed, can be recommended for giving its further consideration. Unfortunately, I cannot recommend the submitted version. The authors are advised to consider the following comments:

1) In the introduction section, explain the scientific and applied novelty of the paper. I think the submitted paper has an applied value and its scientific novelty is very unclear. At the same time, I think all kind of applications of polymers are suitable for publishing in Polymers. Therefore, I think the authors should elaborate more on both scientific and applied aspects of their work.

2) Materials and experimental apparatus. Explain the meaning of “the specifications of ASTM D-150”. At least, some reference to a technical literature should be provided. Specify the area of electrodes. Specify the measured electrical conductivity (AC or DC conductivity).? Explain why the speed of 3 ºC / min was chosen?

3) Discussion. 

Fig.3 quantify the statement “different amount of GTR” by providing an actual weight concentration

Fig. 4 and Fig. 6: The temperature dependence should be explained. What are the mechanisms of the observed nearly temperature-independent conductivity? What are major charge carriers and their basic parameters such as their mobility and concentration?

Figure 5(b) the drop in the value of the imaginary component of the dielectric permittivity around 1 Hz is strange. Is it an error or the manifestation of some dielectric phenomena?

Table 1. The calculated activation energy should be discussed in more detail.

4) Wording must be improved (high properties, in function, one specimen were used, two differentiated behavior, cause LLDPE is,  ).

5) The proposed application (electrical insulators for cables) is actually not supported by the obtained data. Think about something more suitable.

6) Minor comments: improve the quality of Figures, especially Figure 9.

Author Response

Response to Reviewer 2 Comments

The submitted paper reports on the dielectric properties of polymer blends made of low-density polyethylene mixed with various amounts of reused tire powders. The overall goal of the paper is to find possible applications of reused tires. This goal is very appealing and the paper, if properly revised and re-reviewed, can be recommended for giving its further consideration. Unfortunately, I cannot recommend the submitted version. The authors are advised to consider the following comments:

Point 1: In the introduction section, explain the scientific and applied novelty of the paper. I think the submitted paper has an applied value and its scientific novelty is very unclear. At the same time, I think all kind of applications of polymers are suitable for publishing in Polymers. Therefore, I think the authors should elaborate more on both scientific and applied aspects of their work.

Response 1: Thank you for your valuable comments, we are agreeing with the comment on the need for introduction improve and clarify the scientific novelty of the paper. To this end we have modify all the introduction and elaborating scientific and applied aspects from our work. We hope the reviewer will appreciate the changes and improvements in the introduction.

Point 2: Materials and experimental apparatus. Explain the meaning of “the specifications of ASTM D-150”. At least, some reference to a technical literature should be provided. Specify the area of electrodes. Specify the measured electrical conductivity (AC or DC conductivity).? Explain why the speed of 3 ºC / min was chosen?

Response 2. Thank you for your valuable comments and questions. ASTM D 150 is the Standard Test Methods for AC Loss Characteristics and Permittivity (Dielectric Constant) of Solid Electrical Insulation.  American Society for Testing and Materials.

ASTM D-150 describe dielectric measurements methods, according with the ASTM Standard.

Summary of Test Method:

Capacitance and AC resistance measurements are made on a specimen. Relative permittivity is the specimen capacitance divided by a calculated value for the vacuum capacitance (for the same electrode configuration) and is significantly dependent on resolution of error sources. Dissipation factor, generally independent of the specimen geometry, is also calculated from the measured values. These test methods are based upon measuring the specimen capacitance between electrodes and measuring or calculating the vacuum capacitance (or air capacitance for most practical purposes) in the same electrode system. For unguarded two-electrode measurements, the determination of these two values required to compute the permittivity.

Calculation of Permittivity, Dissipation Factor, and Loss Index: The measuring circuits used will give, for the specimen being measured at a given frequency, a value of capacitance and of ac loss expressed as Q, dissipation factor, or series or parallel resistance. When the permittivity is to be calculated from the observed capacitance values, these values must be converted to parallel capacitance. When the parallel substitution method is used, the dissipation factor readings must be multiplied by the ratio of the total circuit capacitance to the capacitance of the specimen or cell. Q and series or parallel resistance also require calculation from the observed values. Permittivity is:  KX = Cp/C v

Cp, a capacitance in parallel with conductance.

Cv, interelectrode capacitance

The properties of dielectric are still the subject of investigation under various operation conditions. As the studies were carried out in different operational conditions and for the different materials, it is hard to make a comparative assessment. For the acceptance of these measurements, some standards are developed (ASTM D-150).

ASTM D-150, References:

[1] K. Ravikumar, K. Palanivelu, K. Ravichandran. Dielectric Properties of Natural Rubber Composites filled with Graphite. Materials Today: Proceedings. Volume 16, Part 2, 2019, Pages 1338-1343. https://doi.org/10.1016/j.matpr.2019.05.233

[2] Melih İnal. Determination of dielectric properties of insulator materials by means of ANFIS: A comparative study. Journal of Materials Processing Technology. Volume 195, Issues 1–3, 1 January 2008, Pages 34-43. doi:10.1016/j.jmatprotec.2007.04.106

[3] A. JadahFarhan, Naajla jerjak. Preparation and Study of Some Physical Properties for Polyester Reinforcement by (Al) Powder Composites. International Journal of Application or Innovation in Engineering & Management (IJAIEM). Vol. 3, Issue 10, October 2014. ISSN 2319 - 4847

[4] D. H. Damon, M. Ezrin, S. Gruchawka. Electrical properties of oven aged cable insulations based on XLPE and EPR. 1983 EIC 6th Electrical/Electronical Insulation Conference, Chicago, IL, USA, 1983, pp. 93-95

[5] S. L. Wang. Comments on the definition of AC capacitance in ASTM-D150-81. Conference on Electrical Insulation and Dielectric Phenomena. Leesburg, VA, USA, 1989, pp. 398-402.

Area of electrodes: 300 mm2, approximately.

According to Jonscher, the electrical conductivity of many disordered solids (including polymer composites) was found to be sum of DC conductivity (independent of frequency) and AC conductivity (strongly frequency dependent). It was noted that the overall frequency dependence of conductivity (the so-called ‘‘universal dynamic response’’ of electron conductivity) could be approximated by equation 1 in paper. 

Where σ0 is the conductivity to direct current (DC),  where f is the frequency, while A and n (which have values between 0 and 1) are parameters which depend on temperature and materials. This equation implies two differentiated behavior, one at low frequencies where DC term is dominant and there is no dependency between conductivity and frequency and another dispersive in which the conductivity has a potential dependence with the frequency, as ω increases the dispersive behavior appears and replaces the DC one. Therefore, as is seen in figure 4a, for low frequencies, we can see the conductivity in direct current (DC) regime, for high frequencies (>100 Hz), and for all the composites, you can see AC current regime.

About why the speed of 3ºC/min was chosen, this heated rate, nearly 3ºC/min, the temperature of the whole sample is homogeneous, according with the materials, and sizes of the specimens. So, this fact is important because the measurement is at the selected or desired temperature.

Point 3: Discussion. 

Fig.3 quantify the statement “different amount of GTR” by providing an actual weight concentration

Response 3: Thank for this comment. When the amount of GTR particles are lower to 20%, dispersion degree improves and the options to avoid the formation of conglomerates also increase. These results agree with the tensile properties obtained in a manuscript published by Yao Dou, Denis Rodrigue “Rotomolding of foamed and unfoamed GTR-LLDPE blends: Mechanical, morphological and physical properties” Cellular Polymers, 2018. 37,2, pag 55-68. All this text has been added to manuscript to clarify your comment.

Point 4: Fig. 4 and Fig. 6: The temperature dependence should be explained. What are the mechanisms of the observed nearly temperature-independent conductivity? What are major charge carriers and their basic parameters such as their mobility and concentration?

Response 4: Thank you for your comment and questions. According with the figure 4b and 6, in the range of temperature 30ºC to 100ºC the electrical is observed nearly temperature-independent conductivity and permittivity. Behavior in figure 4b and 6 is coherent with electrical insulators, the temperature does not affect conductivity process in the analyzed range of temperature (30ºC to 100ºC). Conductive mechanism is described for the temperature range is that when two different charge carriers, such as electrons and holes are separated by a potential barrier, one can move to the other through tunneling the barrier or moving over the barrier via an activated state. This process is called hopping. These phenomena are described in a text in the resubmitted article.

References:

[6] M. J. Billings, A. Smith and R. Wilkins. Tracking in Polymeric Insulation. IEEE Transactions on Electrical Insulation, vol. EI-2, no. 3, pp. 131-137, Dec. 1967.

[7] Shengtao Li, Shihu Yu, Yang Feng. Progress in and prospects for electrical insulating materials. High Voltage, vol. 1, no. 3, pp. 122-129, 2016.

[8] G. Teyssedre and C. Laurent. Charge Transport Modeling in Insulating Polymers: From Molecular to Macroscopic Scale. IEEE Transactions on Dielectrics and Electrical Insulation Vol. 12, No. 5; 2005. DOI: 10.1109/TDEI.2005.1522182

[9] P.S. Davids, I.H. Campbell, D.L. Smith. Device model for single carrier organic diodes. Journal of Applied Physics 82 (12), 6319-6325 (1997); https://doi.org/10.1063/1.366522

Point 5: Figure 5(b) the drop in the value of the imaginary component of the dielectric permittivity around 1 Hz is strange. Is it an error or the manifestation of some dielectric phenomena?

Response 5: Thank you for your comment. This is a measuring error since it only occurs once and for a single value. We have fixed it.

Point 6: Table 1. The calculated activation energy should be discussed in more detail.

Response 6: We would like to thank respected Reviewer for suggestion. The discussion of Ea has been improved in the manuscript and table 2 has been upgraded.

Point 7: 4) Wording must be improved (high properties, in function, one specimen were used, two differentiated behavior, cause LLDPE is,).

Response 7: Thank you for comment. These spelling mistakes have been fixed.

Point 8: 5) The proposed application (electrical insulators for cables) is actually not supported by the obtained data. Think about something more suitable.

Response 8: We are completely agreed with the reviewer, electrical cable application is not suitable. So, in this sense we have discarded the electrical cables application. On the other hand, we have searched low requirements applications such as the selected. Electrical insulating characteristics of the composites LLDPE+GTR worsen with the addition of GTR and therefore it produces insulating mixtures of worse characteristics, with the incorporation of GTR, however it has been found that the analyzed mixtures behave, in general, as good electrical insulators, according to the analyzed standards (Table 3), for low contents of GTR percent’s (5-10-20%), LLDPE+GTR produce materials that have enough properties to constitute electrical insulators for different applications (Conductivity <10-12 S/cm and Tangent of δ <104) since the results analyzed in table 3 in all the samples analyzed with 5-20% GTR has given results well below the limits, as established by the analyzed UNE and IEC standard. As we can check in tables 3 and 4, this give possibilities to use composites with low amounts of GTR for a low requirements applications such as: universal electrical cable joint, filler for electrical applications, cables tray systems, cable ladder systems and pipes for electrical cables, as you can see in table 4.

New tables 3 and 4 are incorporated in paper, in a new section 3.7 Industrial applications analysis: Table 3. Conductivity, loss factor and loss tangent, for each composite, at 50 Hz and 30ºC and Table 4. Applications analyzed according with different standards, and electrical criterion for them. And developments about composites industrial applications, has been introduced in a new section of article 3.7 Industrial applications analysis.

Point 9: 6) Minor comments: improve the quality of Figures, especially Figure 9.

Response 9: According with the reviewer petition, the quality of the figures has been improved. Moreover, all the figures from 4 until 11 have been improved.

Reviewer 3 Report

In this paper, the authors report from SEM detections that it is possible to see GTR particles dispersed in the homogeneous LLDPE polymer matrix. Moreover, the compatibility between GTR and LLDPE is not too good. So, as a result, with high GTR concentrations there is a greater potential for particle agglomeration which impedes interfacial adhesion. The percentage of LLDPE is not enough to wrap the GTR particles, making bonding more difficult, with cracks and pores of considerable size appearing in their contours. The interaction between matrix and reinforcement particles of GTR is very low. As a conclusion, for medium and high concentrations of GTR in the matrix, low levels of linkage appear between components.

-It is necesary to know to which types of composites the reported SEM observations are relative.

-If the most interesting results can be obtained with low GTR contents, why SEM observations of these materials have not been reported and stressed?

-It's important to show a good particle distribution in the matrix in order to justify the electrical properties that have been obtained.

Major revisions are required

Author Response

Response to Reviewer 3 Comments

In this paper, the authors report from SEM detections that it is possible to see GTR particles dispersed in the homogeneous LLDPE polymer matrix. Moreover, the compatibility between GTR and LLDPE is not too good. So, as a result, with high GTR concentrations there is a greater potential for particle agglomeration which impedes interfacial adhesion. The percentage of LLDPE is not enough to wrap the GTR particles, making bonding more difficult, with cracks and pores of considerable size appearing in their contours. The interaction between matrix and reinforcement particles of GTR is very low. As a conclusion, for medium and high concentrations of GTR in the matrix, low levels of linkage appear between components.

Point 1: It is necesary to know to which types of composites the reported SEM observations are relative.

Response 1:  It has been a mistake don’t describe to which sample each SEM picture corresponds. Both pictures are assigned to samples of 20% of GTR. The left picture shows the lack of compatibility between compounds when the GTR particle are large and in the right picture has been observed a homogeneous distribution of GTR particles.

Point 2: If the most interesting results can be obtained with low GTR contents, why SEM observations of these materials have not been reported and stressed?

Response 2: As mentioned in the previous answer, both SEM pictures correspond to the range in which the best properties are obtained. If we use samples with lower GTR content (5, 10%), the amount of GTR particles is so small that it is difficult to obtain good pictures with the SEM. Likewise, the LLDPE is a polymer that under stress always shows this form of rupture with a lot of deformation that leaves a gap around the GTR particle.

Point 3: It's important to show a good particle distribution in the matrix in order to justify the electrical properties that have been obtained.

Response 3: We have added this picture, assigned to a 20% GTR LLDPE sample, which show us a homogeneous distribution of the GTR particles with absence of GTR clusters. Small particles are wrapping for the LLDPE matrix, but in the central part of picture appears a big particle (150 m of length) that explain the lack of interaction between both components. As explain in manufacturing process after mixing process by Brabender equipment has been used an additional mixing with two roll mill to improve the homogeneity of samples. When the amount of particles are higher to 20% the dispersion is difficult but samples with amounts of GTR lower to 20% the degree of dispersion improve.

Point 4: Major revisions are required

Response 4: Thank you for your comment. Completely agreed with the appreciation. Major revision has been performed in the resubmitted article version. We hope you will appreciate the effort in this sense. Moreover, the quality of all the graphs from figure 4 until figure 11 have been improved, and new figures in 3, have been provided, and new references have been added.

Round 2

Reviewer 1 Report

The authors have done a good job in responding to my concerns and comments. The revised version is of a publishable standard. I merely have a minor suggestion - please claim AC or DC conductivity throughout. 

Author Response

Response to Reviewer 1 Comments

Point 1: The authors have done a good job in responding to my concerns and comments. The revised version is of a publishable standard. I merely have a minor suggestion - please claim AC or DC conductivity throughout. 

Response 1: We thank the reviewer’s comments which has leading the way for improving our work.

According to Jonscher, the electrical conductivity of many disordered solids (including polymer composites) was found to be sum of DC conductivity (independent of frequency) and AC conductivity (strongly frequency dependent). It was noted that the overall frequency dependence of conductivity (the so-called ‘‘universal dynamic response’’ of electron conductivity) could be approximated by the following simple relation [1]: 

     or      =  +    ;

Where σ0 is the conductivity to direct current (DC),  where f is the frequency, while A and n (which have values between 0 and 1) are parameters depending on temperature and materials. This equation implies two differentiated behavior, one at low frequencies where DC [2] term is dominant and there is no dependency between conductivity and frequency and another dispersive, in which the conductivity has a potential dependence with the frequency, as ω increases the dispersive behavior appears and replaces the DC one. Taking this into account, and in order to give a satisfactory answer, the measures shown in figure 4a shows the conductivity for a wide range of frequencies (0.01Hz - 3 MHz), and combine two differentiated conductivity regimes: DC (low frequencies) and AC (high frequencies), in low frequency regime, the conductivity measurements domains in DC conductivity regime with addition of higher GTR loads, but for the rest of composites AC conductivity regime prevails at 50 Hz.

References:

[1] Jonscher AK. The universal dielectric Response. Nature 1977; 267: 673–679.

[2] Mahapatra, S. & Vadahanambi, Sridhar & Choudhary, Ram Naresh & Tripathy, D.. (2008). AC conductivity and positive temperature coefficient effect in microcellular EPDM vulcanizates. Polymer Composites. 29. 1125 - 1136. 10.1002/pc.20383.

Reviewer 2 Report

The authors addressed the received comments. The paper can be recommended for its publication.

Author Response

Response to Reviewer 2 Comments

Point 1: The authors addressed the received comments. The paper can be recommended for its publication.

Response 1: We sincerely appreciate the reviewer's comments and to acknowledge the work the team has done to be able to improve the resubmitted paper version.

Reviewer 3 Report

The paper can now be accepted for publication. I suggest to add the error bars in all the figures

Author Response

Response to Reviewer 3 Comments

Point 1: The paper can now be accepted for publication. I suggest to add the error bars in all the figures

Response 1: Thank to reviewer for comments, and acceptation for publication. So few dielectric analysis research works published in research journals refers to the bar errors measurement, therefore most of dielectric characterizations research works do not apply this type error bar [1-3], that is commonly used in other type of analysis such as, for instance, in mechanical characterization research works, therefore will take in account of reviewer suggestion for a future works, in this sense we consider applying the bar error for upcoming papers in dielectric characterization.

Most common sources of errors in dielectric test measurements (DEA)

Errors resulting from the measuring system (e.g., accuracy of the signal analyzer and influence of the cables and sample cell) strongly depend on the frequency of the measurement and the actual sample capacity [4-6]. Such limitations are usually adequately described in instruction manuals supplied by the manufacturers. It is considered that the accuracy is not constant as a function of frequency. It is considered a maximum of error 5% below 0.1 Hz and above 1 kHz about 1% in the range 0.1 Hz − 1 kHz [7]. To minimize the error is needed good electrical contact between sample and electrode plates must be ensured in order to avoid error due to trapped air; an additional capacitance in series with the sample would be introduced, as in our measurements with DEA Equipment. And the expected error due thermal heating was estimated to be in the order of 0.1ºC [8], so it means that we have a percent error thermal heating less than 0,1%, low considered error. In our case DEA test measured error lower than 2%, that is considered with good precision.

References:

[1] Hilker, Brent & Fields, Kimberly & Stern, Abraham & Space, Brian & Zhang, X. & Harmon, Julie. (2010). Dielectric analysis of poly(methyl methacrylate) zinc(II) mono-pinacolborane diphenylporphyrin composites. Polymer. 51. 4790-4805. 10.1016/j.polymer.2010.08.049.

[2] Mahapatra, S. & Vadahanambi, Sridhar & Choudhary, Ram Naresh & Tripathy, D.. (2008). AC conductivity and positive temperature coefficient effect in microcellular EPDM vulcanizates. Polymer Composites. 29. 1125 - 1136. 10.1002/pc.20383.

[3] Guarrotxena, N., Mudarra, M. Influence of polymethylmethacrylate microstructure on its conductive properties at high temperatures. Journal of Nature Science and Sustainable Technology, 13(3), 239-252, 2019.

[4] Vassilikou-Dova, Aglaia & Kalogeras, Ioannis M. (2008). Dielectric analysis (DEA). 10.1002/9780470423837.ch6.

[5] A. Franck. Dielectric Characterization. TA Intruments. September 2012. http://www.tainstruments.com/pdf/literature/APN032%20Dielectric%20characterization_V1_ajf_30SEP12.pdf

[6] Schaumburg, G., & Wilmer, D. Improving the Accuracy of Dielectric Measurements. Novcontrol Technologies. February 2018. https://pdfs.semanticscholar.org/4380/7f1454cdbe54a9c176d16a82ba22ca4d0ca4.pdf

[7] Perez Aparicio, Roberto & Crauste-Thibierge, Caroline & Cottinet, Denis & Tanase, Marius & Metz, Pascal & Bellon, Ludovic & Naert, A. & Ciliberto, Sergio. (2015). Simultaneous and accurate measurement of the dielectric constant at many frequencies spanning a wide range. Review of Scientific Instruments. 86. 044702. 10.1063/1.4916260.

[8] J.H. Wendorff, Th. Fuhrmann. Dielectric spectroscopy and the interaction of light and matter. Dielectrics Newsletter. Scientific newsletter for dielectric spectroscopy. Issue July 1994. https://www.novocontrol.de/newsletter/DNL02.PDF
